# Fabrication, Crystalline Behavior, Mechanical Property and In-Vivo Degradation of Poly(l–lactide) (PLLA)–Magnesium Oxide Whiskers (MgO) Nano Composites Prepared by In-Situ Polymerization

**DOI:** 10.3390/polym11071123

**Published:** 2019-07-02

**Authors:** Hui Liang, Yun Zhao, Jinjun Yang, Xiao Li, Xiaoxian Yang, Yesudass Sasikumar, Zhiyu Zhou, Minfang Chen

**Affiliations:** 1School of Materials Science and Engineering, Tianjin University of Technology, Tianjin 300384, China; 2Chimie des Interactions Plasma-Surface, University of Mons (Umons), 20 Place du Parc, B 7000 Mons, Belgium; 3Key Laboratory of Display Materials and Photoelectric Device (Ministry of Education), Tianjin 300384, China; 4School of Environment Science and Safety Engineering, Tianjin University of Technology, Tianjin 300384, China

**Keywords:** PLLA–MgO whiskers composites, in-situ polymerization, bone repair and fixation, mechanical properties and crystallization behavior, regulating in-vivo degradation

## Abstract

The present work focuses on the preparation of poly(l–lactide)–magnesium oxide whiskers (PLLA–MgO) composites by the in-situ polymerization method for bone repair and implant. PLLA–MgO composites were evaluated using Fourier transform infrared spectroscopy (FTIR), X-ray diffraction (XRD), differential scanning calorimetry (DSC), scanning electron microscopy (SEM) and solid-state ^13^C and ^1^H nuclear magnetic resonance spectroscopy (NMR). It was found that the whiskers were uniformly dispersed in the PLLA matrix through the interfacial interaction bonding between PLLA and MgO; thereby, the MgO whisker was found to be well-distributed in the PLLA matrix, and biocomposites with excellent interface bonding were produced. Notably, the MgO whisker has an effect on the crystallization behavior and mechanical properties; moreover, the in vivo degradation of PLLA–MgO composites could also be adjusted by MgO. These results show that the whisker content of 0.5 wt % and 1.0 wt % exhibited a prominent nucleation effect for the PLLA matrix, and specifically 1.0 wt % MgO was found to benefit the enhanced mechanical properties greatly. In addition, the improvement of the degrading process of the composite illustrated that the MgO whisker can effectively regulate the degradation of the PLLA matrix as well as raise its bioactivity. Hence, these results demonstrated the promising application of PLLA–MgO composite to serve as a biomedical material for bone-related repair.

## 1. Introduction

Biodegradable polymers have been widely utilized as materials for biomedical applications. PLLA has received increasing attention regarding its usage as a biocompatible polymer for various applications, such as for implant materials, surgical sutures, and controlled drug delivery systems. However, there have been some defects including high brittleness, low strength and hydrophobicity and acidic products from its degradation, limiting its practical usage widely [1,2,3]. To overcome these drawbacks, the study of polymer–inorganic composites has attracted great interest since they exhibit greatly enhanced properties. These inorganic fillers with good bioactivity, such as hydroxyapatite (HA) [4,5,6,7], β-tricalcium phosphate (β-TCP) [8] and magnesium oxide (MgO), are beneficial to improving PLA biocompatibility and mechanical properties during the bone repair process. Particularly, the MgO nanowhisker or nanoparticle has attracted interest and has been used for polymer composites, effectively satisfying osteoconductivity in PLLA composites improving cell adsorption and remarkably neutralizing acidic products produced from PLLA degradation [8,9,10,11,12,13,14,15]. Although the surface modification of nano-MgO as a nucleating agent is helpful to promote dispersion and reduce the interface difference of composites [7,16,17,18,19], it can be found that the extra modifying process and types of modifiers frequently cause the cost to increase and require the usage of toxic organic solvents, which cause limitations of its biomedical applications [15,20,21,22,23,24]. Meanwhile, the difference of polarity between hydrophilic inorganic fillers and hydrophobic polymers leads to worse performance regarding biocompatibility and interfacial interaction bonding; additionally, interfacial defects are usually observed in fracture morphology [23,25,26]. For example, Chen prepared PLA/n–MgO composites through blending PLLA and modified MgO [20], but the presented elongation at the break of the composites decreased, apparently due to the agglomeration of modified nanoparticles leading to the reduction of material toughness, probably ascribed to the relatively poor interfacial bonding between the modified n-MgO and PLLA matrix. Urayama reported that the composites of modified PLLA/MgO by oligo–d–lactide and oligo–l–lactide presented as more brittle than the pristine PLLA matrix, and also had the drawbacks of a complex preparation process and higher cost [27].

Undoubtedly, the composites obtained by the in-situ polymerization approach will remarkably reduce costs and promote dispersion through the strong interfacial interaction bonding between fillers and the organic matrix [28,29]. It has been proved that the incorporated fillers, acting as nucleating agents, have an impact on the in-situ ring-opening polymerization process, including properties such as crystallization, mechanical strength and biocompatibility [19,28,29]. For example, Samadi et al. have prepared TiO_2_/polylactide (PLA) composites with different contents of TiO_2_ by the in-situ polymerization method, and the mechanical properties of the synthesized composite were markedly improved [6]. Liu et al. prepared PLA/montmorillonite nanocomposites via the in-situ polymerization of lactide and montmorillonite. As a result, better performance in term of thermal stability and rheological behavior was achieved [30]. Li et al. prepared PLLA–MgO nanocomposites with a weight-average molecular weight of 55,500 by in-situ melt polycondensation from l–lactic acid and surface-hydroxylated MgO, in spite of the nanocomposite achieving better mechanical properties than pristine PLLA; however, MgO still needed be modified, and the biocompatibility of the synthesized composite also has not been investigated [24]. To the best of our knowledge, there have been few reports on the in-situ preparation of PLLA–MgO composite via an in-situ solution reaction combined with the melted reaction.

Here, a simple and low-cost technique was explored to prepare the PLLA–MgO composite by the in-situ polymerization of l–lactide on the surface of an MgO whisker for bone repair and fixation applications. The bulk composites, in terms of their structure, molecular weight and interfacial morphology, were systematically investigated by different characterizations. Meanwhile, the effect of the MgO whisker on regulating the degradation of PLLA was also evaluated through the in vivo experiments.

## 2. Experimental Materials

Powdered l–lactide was purchased from Daigang Biological Engineering Co., Ltd. (Jinan, China). Methyl alcohol (>99.9% HPLC) and chloroform (>99.8%) were purchased from Aladdin (Brøndby, Denmark). NaCO_3_, MgCl_2_·6H_2_O and methanol (>99.9%) were supplied by Ji Zhun Chemical Co., Ltd. (Tianjin, China). l–lactide was recrystallized three times by toluene prior to the synthesis, and then dried under vacuum at 60 °C for 6 h. All commercial chemicals were of analytical-grade purity.

### 2.1. Preparation of Magnesium Oxide Whiskers

MgO whiskers were prepared in the laboratory [20]. Briefly, 100 mL of Na_2_CO_3_ (0.6 mol/L) was added dropwise into an equal volume of MgCl_2_ (0.6 mol/L) and stirred for 20 min. The mixture was aged at room temperature for 10 h and then filtered, washed, and dried at 80 °C for 3–4 h. The precursor was calcined at 750 °C for 4 h with a heating rate of 5 °C/min. Then, 1.92 g of the resultant MgO whiskers were obtained with a yield of 80%, a length of 50 μm and a radial size of 200 nm.

### 2.2. In-Situ Polymerized PLLA–MgO Composites

Firstly, a total of 10 g l–lactide and MgO whisker were dried in vacuum oven at 50–70 °C for 60 min. After that, the mixture of l–lactide, MgO whisker, Sn(oct)_2_ and chloroform were added into a 100 mL three-neck flask and placed in an ultrasonic bath to achieve a full dispersion for about 30 min. Subsequently, the flask was equipped with a mechanical stirrer and a reflux condenser was connected with a vacuum system, and then the flask was refluxed in oil bath at 80 °C for 6 h. Next, the temperature was gradually increased to 145 °C and stirring performed for 10 h. After the solvent was extracted, the mixture was melted and underwent reaction for about 24 h. At the end of the reaction, the flask was cooled, and the product was dissolved in chloroform and subsequently precipitated into methanol to eliminate unreacted l–lactide. The resulting solid was filtered and dried under vacuum at 80 °C for 24 h. The obtained samples were marked according to their MgO loading ratio as PLLA–0 wt % (PLLA), PLLA–0.5 wt % MgO (PLLA0.5), PLLA–1.0 wt % MgO (PLLA1), and PLLA–1.5 wt % MgO (PLLA1.5). The detailed information of the samples is given in Table 1. In addition, the composite films were prepared for future characterization and analysis, and the preparation procedure is shown in the Appendix A.

### 2.3. Molecular Weight (M_η_) Measurements

Molecular weight (*M*_η_) measurements were carried out to test the *M*_η_ of as-synthesized samples (Table 1) and the samples after in vivo degradation at each period (Table 2). The viscosity-average molecular weights (*M*_η_) of the samples were identified by viscometric measurements using an Ubbelohde Capillary Viscometer, type 1835 (0.3–0.4 μm). The value was calculated with the [η] = K*M*_η_ α equation, where K = 11.2 × 10^−4^ (dm^3^/g) and α = 0.73, determined in chloroform at 25 °C. The measured solution containing samples should be filtered using a hydrophilic membrane filter with a size of 0.45 microns before the analysis. The detailed description of the measurement process of molecular weight (*M*_η_) is shown in the Appendix A.

### 2.4. Characterization

X-ray powder diffraction (XRD, Rigaku D/max/2500PC, Tokyo, Japan) was performed using Cu Kα radiation with λ = 1.5418 Å, operating at 40 kV/100 mA, with a scanning speed of 8°/min. Fourier transform infrared (FTIR) spectra were obtained from KBr pellets using a Bruker tensor 37 spectrometer (Bruker, Billerica, MA, USA) in the range of 4000–400 cm^−1^, with a spectral resolution of 4 cm^−1^ and an average of 32 scans. The thermal analysis of samples was performed using a DSC instrument (Netzsch Co. Ltd., Freistaat, Germany). The temperature and heat flow were calibrated using an indium standard under nitrogen purging; the samples (5–8 mg) were weighed and sealed in an aluminum pan and heated under nitrogen flow from room temperature to 220 °C at a heating rate of 20 °C/min. The samples were cooled from 200 to 20 °C at 5 °C/min for non-isothermal crystallization. For the second crystallization, the samples were reheated to 220 °C at a heating rate of 10 °C/min. The crystallinity degree (*X_c_*) of samples was estimated using the following equation:$$X_{c}=\frac{\Delta H_{m}+\Delta H_{rc}}{\Delta H_{m}^{\infty }\times \phi }\times 100\%$$
where Δ*H_m_* (J/g) is the value of fusion, *H_rc_* is the recrystallization enthalpy obtained during the DSC heating process, and Δ*H_rc_* is the fusion enthalpy of the completely crystalline PLLA, and *ϕ* is the weight fraction of PLLA in the sample. The value of PLLA is selected as $\Delta H_{m}^{\infty }$ = 93.6 J/g [30].

The chemical structure was investigated by means of proton nuclear magnetic resonance (^1^H and ^13^C NMR) with a Varian Mercury Plus NMR 400 MHz apparatus. Chemical shifts (*δ*) in ppm were assigned to the residual solvent proton at *δ* = 7.26 ppm (Appendix A). The samples used in mechanical testing were fabricated into dumbbell shapes as shown in Figure 1A, and the tests were conducted using an electromechanical universal testing machine (Instron, Norwood, MA, USA) with a loading rate of 0.1 mm/min. The preparation process of the standard dumbbell specimen is shown in the Appendix A; these standard dumbbell specimens were used in the mechanical experiment, and three samples were tested for each experiment. The morphology of the samples was characterized by field-emission scanning electron microscopy (FESEM, JOEL 6700F, Japan, operating at 10 KV).

### 2.5. In Vivo Experimental

In Figure 1B, the PLLA and PLLA1 rods (φ2 mm × 6 mm) were prepared by a micro extruder (Wuhan Ruiming Test Equipment, Ltd., Wuhan, China) in advance, which were used for in vivo degradation tests. The injection temperature of the samples was 190 °C.

### 2.6. Animal Models

The experimental protocol was approved and implemented by the Animal Care and Use Committee of Tianjin Hospital and performed in strict accordance with the recommendations from the Guide for Animal Management Rules and international ethical normative from the Ministry of Health of the Peoples Republic of China 2001/545/China (approval No. 2015-11154). In this study, a total of 10 healthy adult (~1 year) Japanese white rabbits weighing 3 ± 0.2 kg were selected for animal testing, and they were divided in 2 groups, with 5 rabbits in one group, of which two rabbits were used as standby samples. One rabbit was implanted with 2 samples, where the left and right legs were implanted with one sample each (φ2 mm × 6 mm). Further, sub-cage feeding was performed for a week and no adverse reactions were found. Rabbits were anesthetized with an intramuscular injection of ketamine (0.2 mL/kg). After that, the hair on the side of the knee in a roughly 5 cm range was shaved. The iodine disinfectant was used to disinfect the knee parts of the rabbit. The anterior lateral patella of the knee was incised to about 4 cm, followed by cutting the skin, lateral support and a joint capsule. A hole with a depth of 1 cm was drilled in the femur bone of the knee by Kirschner wire drilling (φ2 mm) for six rabbits, and the other two were drilled in both the femur and tibial cancellous bone of the knee. The PLLA and PLLA1 rods were implanted into the hole separately; then, postoperative suture, iodophor disinfection and sub-cage feeding were performed. The implantation process is shown in Figure 1B. Rabbits were sacrificed with ear veins injected with air. The bone with the implanted rod was removed from the euthanized rabbits after 4, 8, 12 and 24 weeks, and preserved by 10% formalin solution. The serial numbers of the implanted rods are shown in Table 2. For all the animal experiments, the materials and surgical instruments were radiation-disinfected in Tianjin Jinpeng far radiation Co., Ltd. after Co_60_ for 24 h, with a radiation dose of 25 KGy.

### 2.7. Routine Pathological Examinations

Hard histological biopsies were performed in the third and sixth month to evaluate the structure variation of implants under long-term degradation behaviors and the tibial cancellous bone response after surgery. The surgical sites were fixed in 10% formaldehyde solution, and then the samples were dehydrated in order of the graded series of alcohols. Following dehydration and decalcification, the specimens were embedded in paraffin, and tissue sections were stained with hematoxylin and red staining.

The implanted PLLA and PLLA1 in the third month and the sixth month were taken from the femur bone, and the *M*_η_ values were measured by a Ukrainian-style viscometer. Each sample was divided into three parts for the molecular weight (*M*_η_) measurements. SEM was used to observe the morphology of the implanted rods during the degrading process.

## 3. Results and Discussion

### 3.1. Molecular Weight

The average molecular weights of PLLA, PLLA0.5, PLLA1 and PLLA1.5 were 36 ± 1.0 (×10^4^), 34.3 ± 0.8 (×10^4^), 34.6 ± 1.3 (×10^4^) and 26.3 ± 1 (×10^4^), respectively, and are shown in Table 1. It can be seen that the *M*_η_ of PLLA is higher than the nancomposite, which is probably due to several polymer chains of nanocomposite growing on the surface of the MgO nanowhisker, which led to stearic hindrance from neighboring polymer chains. The *M*_η_*s* of PLLA0.5 and PLLA1 were higher than PLLA1.5, which was attributed to more MgO aggregating in the PLLA matrix and a decrease in the mobility of polymer molecules when compared with a single free polymer, preventing the growth of the chain [9]. However, the chains of PLLA1 achieved a larger hydrodynamic volume at the optimal MgO loading of 1.0 wt % on the surface of the MgO whisker, leading to a higher molecular weight than PLLA0.5 and PLLA1.5. Here, PLLA1 were chosen as the experiment sample for the characterization studies.

### 3.2. X-Ray Diffraction

The XRD patterns of MgO, PLLA and PLLA1 are shown in Figure 2. As compared to the diffraction peak of pure PLLA, the diffraction peaks at 37°, 62.44°, 74.76° and 78.44° in PLLA1 were moved from 37.08°, 62.3°, 74.84° and 78.76°, respectively, and are assigned to the MgO whisker and the weakened MgO peaks because of their lower abundance [31]. The peaks at 16.72°, 19°, 20.52° and 22° in PLLA1 were shifted to 16.74°, 19.12°, 20.96° and 22.24°, respectively. These changes were observed previously due to the chemical interaction between PLLA and MgO whiskers, and the interaction was further analyzed by FTIR spectroscopy.

### 3.3. Fourier Transform Infrared Spectroscopy

The FTIR analysis is performed to obtain qualitative estimations of the changes of functional groups of PLLA1 compared to neat PLLA. Figure 3 shows the IR spectra of MgO, PLLA and PLLA1 composites, respectively. The absorption bands at 2951 cm^−1^ in the PLLA1 correspond to the stretching of the –CH– groups in the main chain of PLLA [32]; compared to the value of 2994 cm^−1^ of PLLA, a small shift in PLLA1 was ascribed to the asymmetric vibration of long alkyl chains of PLLA on the surface of MgO. The peak of the C=O stretching vibration of the carbonyl groups in PLLA were shifted to a higher wavenumber in the region from 1760 to 1754 cm^−1^, which was caused by the chemical bond between the COOH of PLLA and OH of MgO [20]. As illustrated in Appendix A, the ^1^H NMR and ^13^C NMR spectra of PLLA1 represent the CH peak at 1.57 ppm, which was split into three distinct peaks and, differently from PLLA, was attributed to the formation of intermolecular H-bonding, as well as the charge-transfer interaction, while the –OH or COOH groups were associated with CH_3_, CH, and C=O groups were not observed due to the impact of the relatively longer chain, leading to the intensity of these peaks decreasing dramatically. As per these results, the shifts of peaks have several differences from the reported blended and surface-modified composites [20].

### 3.4. Scanning Electron Microscopy

The surface morphology of PLLA, PLLA1 and MgO whiskers were analyzed by the SEM measurements and are shown in Figure 4. The nanocomposites had a continuous morphology with small, uniformly distributed convexities compared to pure PLLA (Figure 4a), indicating that MgO whiskers were uniformly dispersed in the matrix, attributed to the formation of H-bonding between organic PLLA chains and MgO, further improving the interfacial interaction. Meanwhile, it was observed that the MgO whiskers were tightly and evenly coated with PLLA, with almost no cavities. In Figure 5b–d, the element distribution shows that MgO rods were well-distributed in PLLA and no byproducts were present, whereas the PLLA chains were in-situ polymerized on the surface of MgO, contributing to the formation of a strong interfacial interaction between the inorganic and the organic phases, which aided in the inhibition of the compact aggregation of MgO whiskers in the PLLA matrix [33].

### 3.5. Synthesized Mechanism of PLLA–MgO Composite

From the characterization results, the in-situ polymerization of l–lactide on the surface of MgO whiskers was used, which is an effective, inexpensive and achievable method for the synthesis of PLLA–MgO composite. The synthesizing mechanism of PLLA–MgO composite is shown in Figure 6. The MgO whisker was attracted to the oligomer, absorbing –OH groups, which was generated through the protonation of surface oxide ions, then bonded to several Mg^2+^ in the solution reaction procedure. During the melted reaction process, PLLA was in-situ polymerized to the surface of MgO by mixing with the carboxyl groups. Due to the interaction and bonding force between the MgO and molecular chain of PLLA, the composite would have avoided external stress and maintained high strength, exhibiting prominent performance.

### 3.6. Crystallization Property of PLLA–MgO Composites

For the secondary crystallization (Figure 7), the melting temperature (*T*_m_) of PLLA appeared at 176.5 °C, which was lower than *T*_m_ (177 °C) of PLLA0.5, while PLLA1.0 and PLLA1.5 appeared at 175.3 and 174.8 °C, respectively, even lower than PLLA. On the other hand, as the amount of MgO whisker increased, the cold crystallization (*T*_cc_) of PLLA1 and PLLA1.5 were increased slightly compared to that of PLLA; however, the cold crystallization temperature (*T*_cc_) of PLLA0.5 was lower than pure PLLA, suggesting PLLA with a 0.5 wt % MgO whisker could cold crystallize faster than others. Referring to the DSC analysis results for *X*_c_ in Table 3, the composites were easier to crystallize during the cooling process, suggesting that the effect of the whiskers as nucleating agents was quite prominent, especially for PLLA0.5, which has a higher *X*_c_ than others. All of these results suggest an enhanced crystallization ability of PLLA in the presence of MgO whiskers. Although the amount of MgO increased to 1.5 wt %, the crystallinity degree of PLLA was decreased, and it was still higher than the crystallinity degree of pure PLLA, indicating that an excess of whiskers is not beneficial to *X*_c_, which was consistent with our reported results [20,33].

In order to fully understand the effect of MgO whiskers on the crystallization of PLLA, the isothermal crystallization behaviors of pure PLLA and nanocomposites with various MgO contents were investigated and are shown in Figure 8. For the same sample, as the isothermal crystallization temperature increased from 100 to 130 °C, the crystallization peak shifted to a longer time. For the same isothermal crystallization temperature, the nanocomposite exhibited a shorter crystallization time compared to pure PLLA; in particular, PLLA0.5 took the shortest time to complete its crystallization, indicating the MgO whisker as a nucleating agent can effectively improve the crystallization of PLLA. However, as the MgO content increased, the crystallization time of PLLA increased, because 1.5 wt % MgO retards the crystallization of PLLA, leading to the crystallization time of PLLA1.5 being even longer than that of pure PLLA, demonstrating that excessive MgO would be aggregated in the PLLA matrix and hinder the movement of the PLLA molecular chain to reinforce stereo-hindrance. In addition, referring to our reported article, it was also proved that an excess of whiskers can produce poor crystallinity in composites due to whisker aggregation, leading to cavities and interfacial defects [33].

### 3.7. Mechanical Properties of PLLA–MgO Composites

Figure 9 and Figure 10 show that MgO has an effect on the improvement in terms of the tensile strength, elastic modulus and elongation at breaks in nanocomposites when compared to pure PLLA. Four samples were tested for one content of MgO. As shown in Figure 10, the tensile strength, elastic modulus and elongation at breaks in PLLA0.5 and PLLA1 initially had a rising tendency; the tensile strength of PLLA1 increased by 68% compared to pure PLLA, the elongation at breaks of PLLA1 was raised by 16.3 times, and the Young’s modulus of PLLA1 was 1.64 GPa, higher than the values of PLLA (611 MPa). Hence, at an MgO loading ratio of 1.0 wt % or less, the mechanical properties of PLLA were significantly improved because the stress concentration could be avoided, which was helpful in strengthening the mechanical properties by inhibiting crack propagation [34,35,36]. As the amount of MgO whisker was increased to greater than 1.0 wt %, the MgO would be aggregated in the PLLA matrix, decreasing the mobility of the PLLA molecular chain, leading to the crack propagation of PLLA being aggravated, and meanwhile the mechanical properties including the tensile strength, Young’s modulus and elongation were decreased. These results were consistent with the DSC analysis, wherein less MgO had an positive effect on the crystalline structure of PLLA, while excessive MgO whiskers hindered the movement of the PLLA molecular chain. Moreover, our group already studied the effect of MgO whiskers on the mechanical properties and crystallization behavior of PLLA in a previously reported article [20], as well as the SEM images of fracture morphologies and surface morphologies of samples having been shown in that article. When the MgO whisker amount was greater than 1 wt %, the deformation of the nanocomposite became more pronounced, leading to a serious interfacial separation phase and defects in the matrix; in particular, the MgO whiskers of 2 wt % and 3 wt % were obviously aggregated in the PLLA matrix. These results could explain why PLLA1.5 had the poorest mechanical properties.

### 3.8. In Vivo Degradation of PLLA and PLLA1

In this study, based upon the aforementioned results for the mechanical properties of the composites, PLLA1 had higher superior tensile strength and was chosen as the experimental material for in vivo degradation experiments.

The variations of molecular weight (*M*_η_) for the specimen in different periods are shown in Figure 11. It was found that there was a slight difference in the molecular weight for PLLA and PLLA1 for a period of 2 months. Specifically, the PLLA1 possessed a relatively higher *M*_η_ before 8 weeks, indicating that the PLLA1 had a slower degrading rate due to the higher crystallinity degrees provided by MgO whiskers from PLLA1. Also, more tightening and a compact area in the crystalline region was more beneficial to delaying the decomposition of PLLA and decreased its hydrolysis rate by preventing the entrance of water [15,18]. Meanwhile, MgO was able to increase the value of pH after dissolving, and the performance of PLLA1 was influenced by MgO in the implanting time of the first and second month, which could neutralize the acid produced from the decomposition of PLLA and inhibit its autocatalytic behavior. Then, a remarkable discrepancy with a sharp decline of *M*_η_ was observed for the third and sixth month, suggesting the occurrence of rapid decomposition for PLLA1. The reversal of the degrading rate in PLLA1 probably resulted from the increment of the pH value due to the dissolution of more hydrophilic MgO whiskers in relatively long-term degradation, which accelerated the hydrolysis of PLLA since it had a higher decomposing rate in an alkaline environment [37].

It can be observed from the SEM images of the rods’ surfaces (Figure 12) that there were no remarkable differences between PLLA and PLLA1 as shown in C1 and E1 during a short testing period. The differences observed in C1 and E1 were probably caused by the adhesion of bone organization when taken out from the bone cavity since the matrix had good biocompatibility with bone. From Figure 12 (C2,E2), the surface of the pure PLLA and the PLLA1 composite were a little tougher, and in particular, there was a large hole observed in C2, probably ascribed to the unconsciously contrived destruction in the removal procedure, which also implied the gradual weakening and collapsing of the matrix as a result of degradation. Nevertheless, it can be seen that the surface of C3 was slightly smoother than that of E3 and, with time, the morphology of E4 was much tougher, even displaying a hole in the bulk matrix. The results revealed that the composite was degraded more intensively after 8 months which consisted of the variation of the *M*_η_ of the samples, as mentioned above. A number of research works have focused on the in vivo degradation of PLA and its composites for a comparatively short period. More attention has been given to the variation of the materials in vivo with a longer implanting time (the 3 and 6 months), and the graphs of histological examinations stained by hematoxylin and red staining are displayed in Figure 13. After 3 months, the edge of the implanted PLLA1 composite was not flat, while the control PLLA was still relatively smooth in Figure 13A, implying that the extent of the decomposition of the composite was more intensive. Moreover, after six months of implantation, it was notable that the implant in E4 exhibited a decomposed crack, even when the apparent break of the implant was observed (E4 of Figure 13B), while the edge status of the PLLA sample displayed in C4 exhibited a similar degradation to the PLLA1 in E3, as seen in Figure 13B. It can be suggested that the PLLA1 with an MgO whisker had a faster degrading rate than that of pure PLLA. Meanwhile, from Figure 13B, it can be seen that the amount of bone cell in the bone formation tissue was slightly different three months later. Specifically, more bone cells were present on the implanted PLLA1, and this could be attributed to the better bioactivity of the composite supported by MgO whiskers, because of the significant effect of Mg^2+^ on bone formation and healing [35,37]. These aforementioned results demonstrate that the degrading rate of the composite changed significantly, which proved the positive effect of MgO whiskers in regulating and controlling the PLLA degradation to improve the bioactivity of PLLA for facilitating bone repair.

## 4. Conclusions

PLLA–MgO nanocomposites were successfully fabricated via the in-situ polymerization of l–lactide and whiskers. The PLLA–MgO composites exhibited a tight interfacial bonding between MgO whiskers and the PLLA matrix. Amounts of 0.5 wt % and 1.0 wt % MgO can apparently promote the crystallization of the PLLA matrix, and even when the amount of MgO whisker was increased to 1.5 wt %, the crystallinity of PLLA was still higher than pure PLLA. Particularly, when using MgO with a loading ratio of 1.0 wt %, the mechanical properties were significantly improved, since MgO whiskers inhibit the crack propagation of PLLA. Moreover, MgO whiskers were able to regulate and control the degradation of PLLA effectively, presenting a slower degradation rate before 8 weeks and rapid decomposition in the following process. These PLLA–MgO nanocomposites can act as a biomedical material with potential application in bone-related repair.

## Figures and Tables

**Figure 1 polymers-11-01123-f001:**
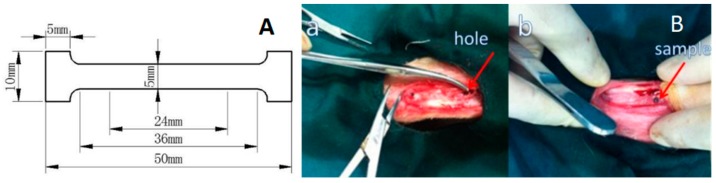
Tensile sample (**A**) and images of PLLA and PLLA1 rods (**B** (**a**)) before and (**B** (**b**)) after implantation in rabbit femur condyles.

**Figure 2 polymers-11-01123-f002:**
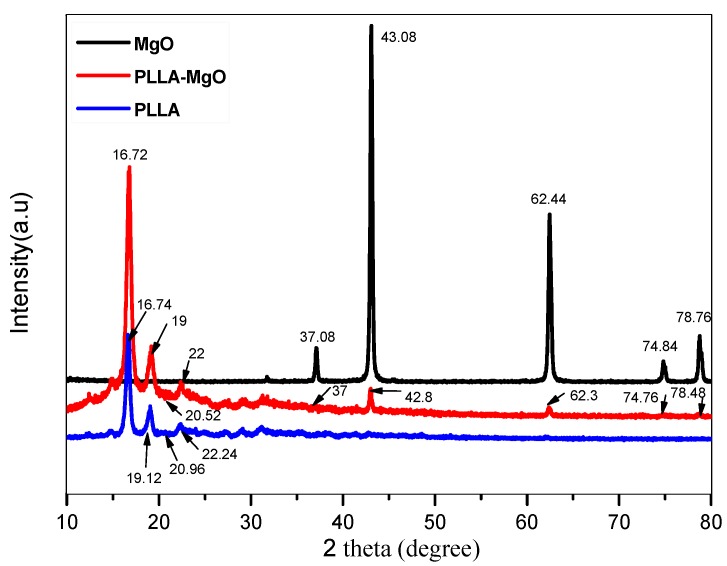
The X-ray diffraction pattern of MgO, PLLA, and PLLA1 composites.

**Figure 3 polymers-11-01123-f003:**
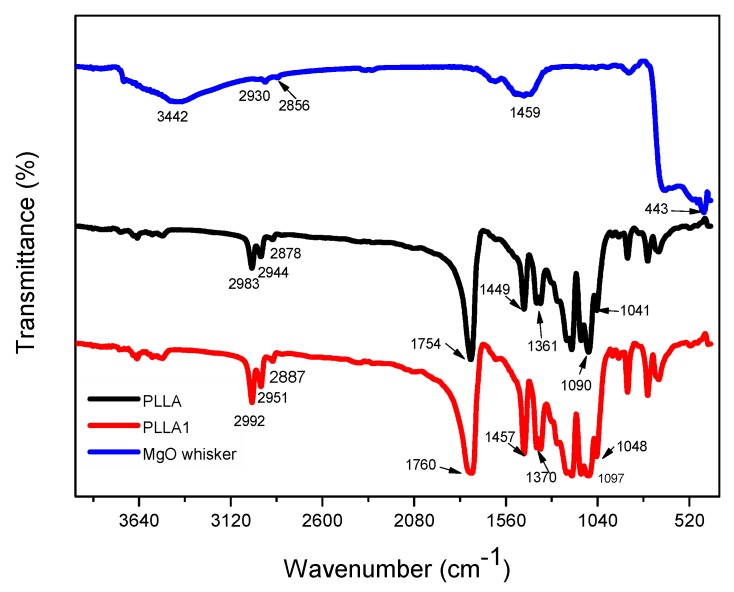
FTIR spectra for PLLA, MgO and PLLA1.

**Figure 4 polymers-11-01123-f004:**
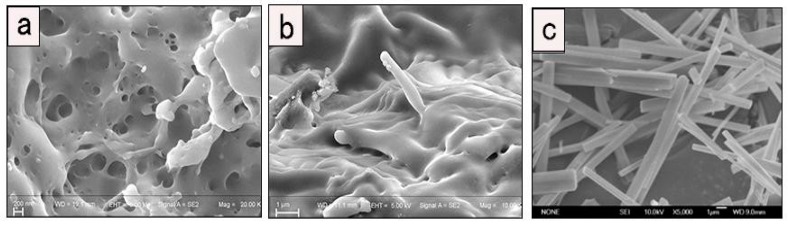
SEM images of PLLA (**a**), PLLA1 (**b**) and MgO whiskers (**c**).

**Figure 5 polymers-11-01123-f005:**
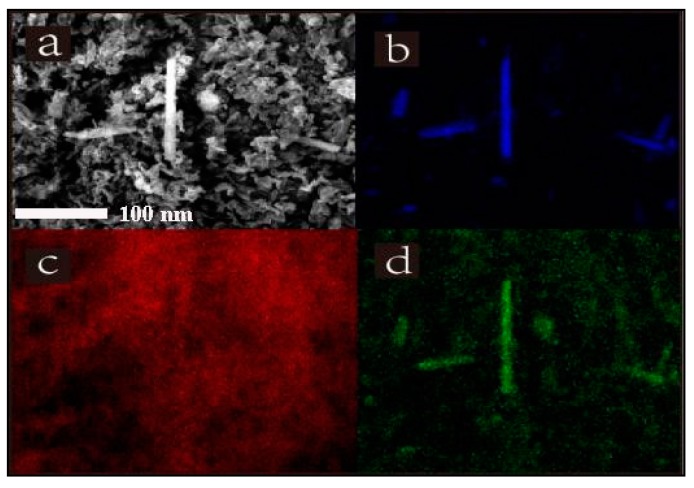
SEM image of the PLLA1 (**a**) and element distribution ((**b**) oxygen, (**c**) carbon, (**d**) magnesium).

**Figure 6 polymers-11-01123-f006:**
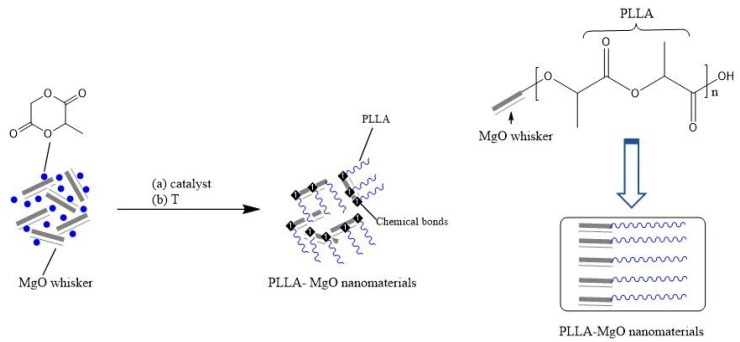
Schematic mechanism of PLLA–MgO via the illustration of chemical bonding between MgO and the PLLA matrix ((a) catalyst, (b) temperature).

**Figure 7 polymers-11-01123-f007:**
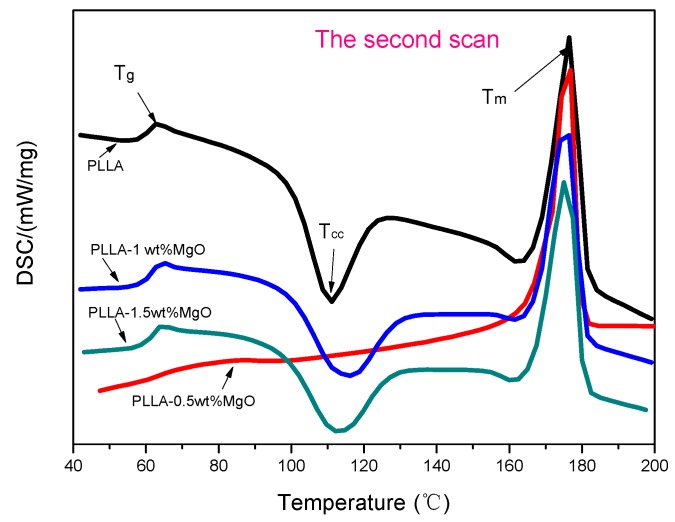
Differential scanning colorimetry (DSC) traces of PLLA, PLLA0.5, PLLA1 and PLLA1.5 for the secondary crystallization: the cooling rates are 5 °C/min and the heating rates are 10 °C/min.

**Figure 8 polymers-11-01123-f008:**
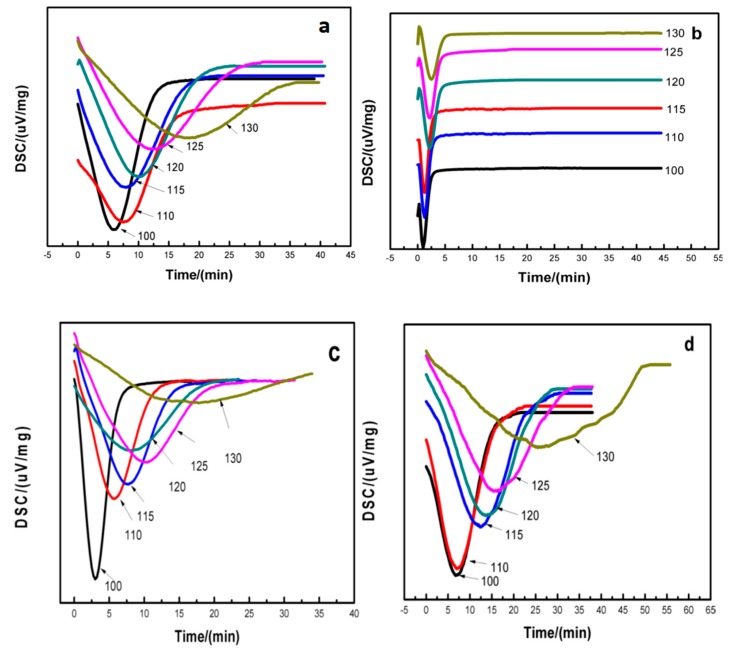
DSC curves of (**a**) pure PLLA, (**b**) PLLA0.5, (**c**) PLLA1.0, and (**d**) PLLA1.5 in the temperature range from 100 to 130 °C, respectively.

**Figure 9 polymers-11-01123-f009:**
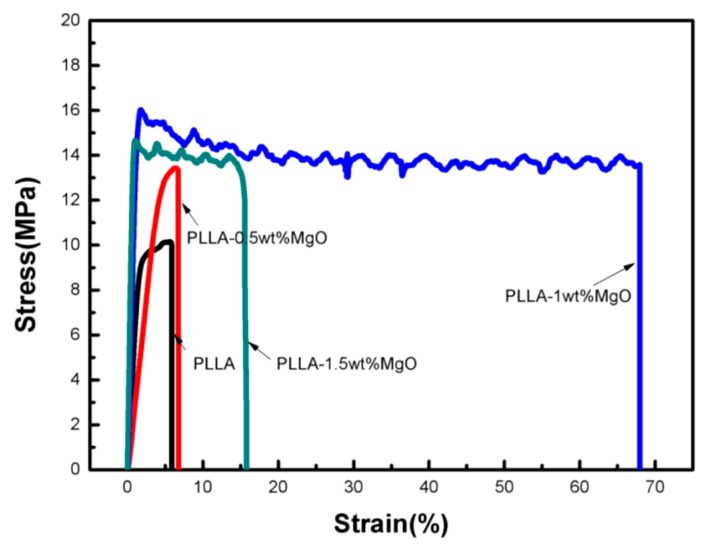
Mechanical properties of PLLA and PLLA–MgO composites.

**Figure 10 polymers-11-01123-f010:**
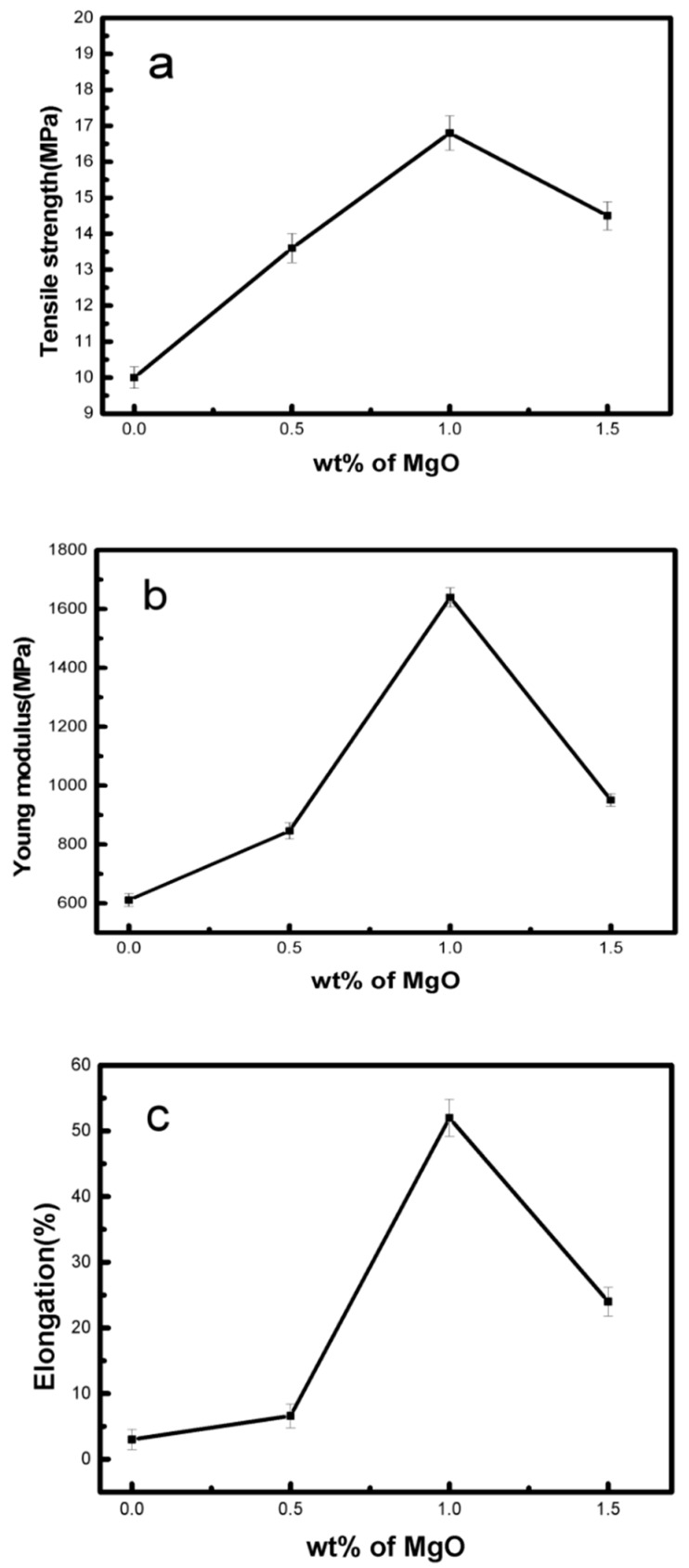
Tensile strength (**a**), Young’s modulus (**b**) and elongation (**c**) of neat PLLA, PLLA0.5, PLLA1 and PLLA1.5.

**Figure 11 polymers-11-01123-f011:**
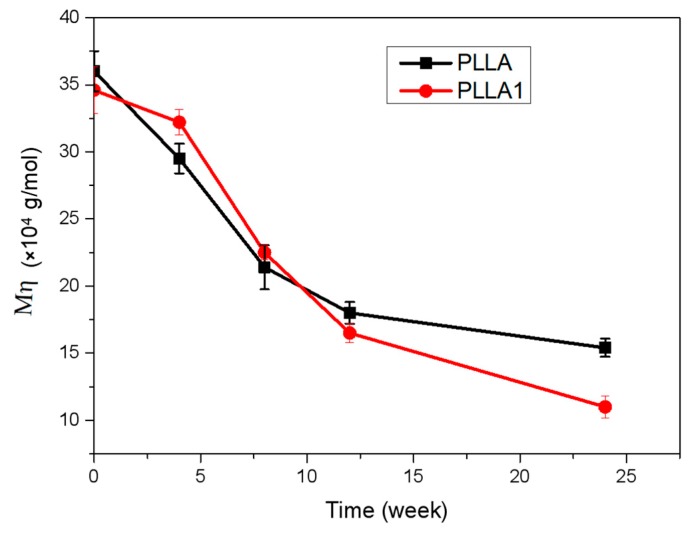
The weight average molecular weight (*M*_η_) of PLLA and PLLA1 in the rabbit body.

**Figure 12 polymers-11-01123-f012:**
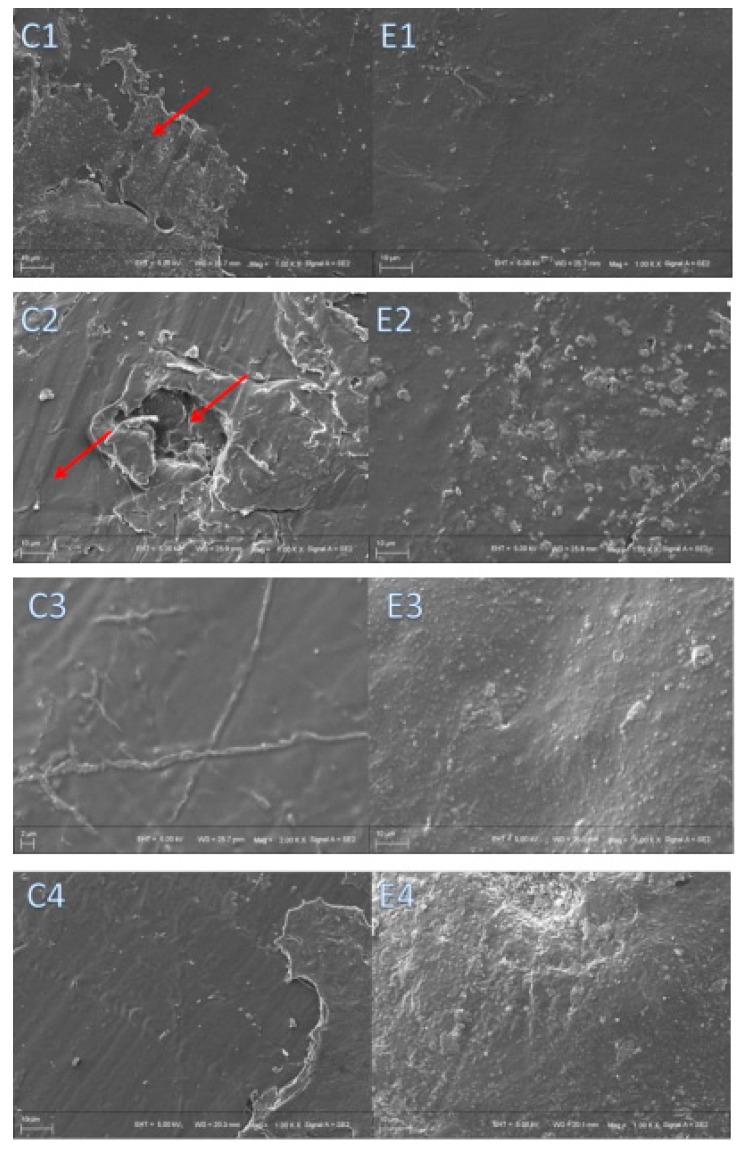
SEM images of the PLLA rod ((**C1**), (**C2**), (**C3**), (**C4**)) and PLLA1 ((**E2**), (**E2**), (**E3**), (**E4**)) rod after implantation for 4, 8, 12 and 24 weeks.

**Figure 13 polymers-11-01123-f013:**
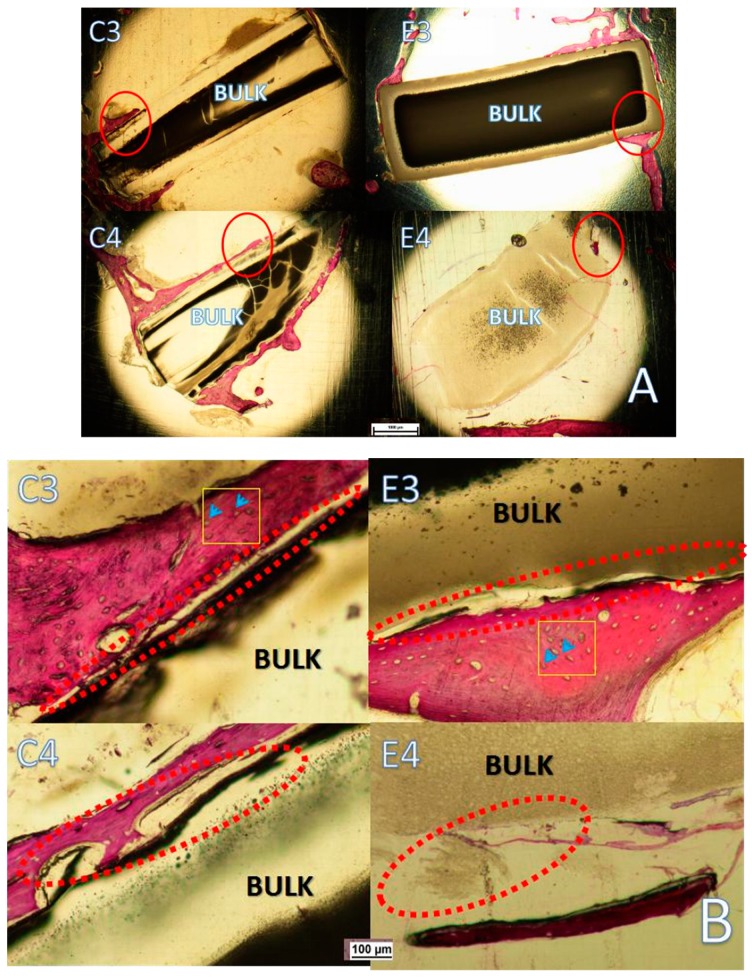
Histological morphologies of implanted PLLA and PLLA1 after 3 and 6-month durations of implantation: (**A**) The whole samples; (**B**) the partial enlargement; the blue arrow indicates the bone cell.

**Table 1 polymers-11-01123-t001:** Detailed information of the samples. PLLA: poly(l–lactide).

Abbreviation of Samples	Sample	Weight of Lactide (g)	Weight of MgO Whisker (g)	Molecular Weight (×10^4^)	Weight of Sn(oct)_2_ (g)
PLLA	PLLA	10	0	36.0 ± 1.0	1.0
PLLA0.5	PLLA–0.5 wt% MgO	10	0.05	34.3 ± 0.8	1.0
PLLA1	PLLA–1.0 wt% MgO	10	0.1	34.6 ± 1.3	1.0
PLLA1.5	PLLA–1.5 wt% MgO	10	0.15	26.3 ± 1	1.0

**Table 2 polymers-11-01123-t002:** Serial numbers of PLLA and PLLA1.

Implantation Time (Months)	PLLA	PLLA1	Molecular Weight (×10^4^)	
			PLLA	PLLA1
1	C1	E1	29 ± 1	32.5 ± 1
2	C2	E2	21 ± 1.3	23 ± 0.8
3	C3	E3	18 ± 0.9	16 ± 0.6
6	C4	E4	16 ± 0.5	11.3 ± 0.5

**Table 3 polymers-11-01123-t003:** The non-isothermal melt crystallization parameters of pure PLLA and composites.

Sample	Δ*H_m_* (J/g)	Δ*H_rc_* (J/g)	*X_c_* (%)	*T*_g_ (°C)	*T*_m_ (°C)	*T*_cc_ (°C)
PLLA	65.63	−40.56	27.6	63.3	176.5	110.8
PLLA0.5	68.09	−3.3	69.3	67.6	177	101.5
PLLA1	67.80	−42.54	37.6	64.7	175.3	115
PLLA1.5	66.94	−44.13	34.6	64.2	174.8	113

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
