# Peer review of "Fabrication, Crystalline Behavior, Mechanical Property and In-Vivo Degradation of Poly(l–lactide) (PLLA)–Magnesium Oxide Whiskers (MgO) Nano Composites Prepared by In-Situ Polymerization"

_polymers, 2019, doi:10.3390/polym11071123_

Round 1

Reviewer 1 Report

The manuscript entitled "Fabrication, crystalline behavior, mechanical property and in-vivo degradation of Poly(L-lactide) (PLLA) -magnesium oxide whiskers (MgO) nano composites prepared by in-situ polymerization" reports about the formulation of PLLA-based composites, containing different contents of MgO nanowiskers, and their characterization. 

In my opinion, the paper needs to be intensively revised before to be suitable for publication in Polymers.

Particularly, the following concerns have to be solved:

- The English language of the whole manuscript needs to be improved.

- The references that appear in the text are not in the right order; as an exemple, in page 2 (Introduction part) ref. 26 appears before than ref. 17.

- Concerning DSC measurements, the Authors stated that the introduction of MgO nanoparticles causes an increase of the PLLA crystallinity. To confirm this finding, the crystallinity of the samples has to be calculated.

- In the comment about the mechanical properties, the worsening of the mechanical performance of the composite containing 1.5 % of MgO was attributed to agglomeration phenomena of the nanoparticles. This finding should be probed through morphological analyses.

Author Response

We would like to thank the reviewers for their detailed checking and helpful comments for our manuscript. The major improvement has been carried out in the revised manuscript.

Reviewer 2 Report

The author answers almost all my concerns. However, In the M&M the author only stated that "The experimental protocol and the approval number of the animal experiment were approved by the Tianjin Hospital." Please provide the approved document number in the text. 

Author Response

(The authors gave the same response as above.)

Round 2

Reviewer 1 Report

The authors have intensively revised the manuscript, according to the suggestions of the Reviewers, therefore I recommend the publication of the manuscript although the English language needs further improvements.

This manuscript is a resubmission of an earlier submission. The following is a list of the peer review reports and author responses from that submission.

Round 1

Reviewer 1 Report

The manuscript entitled "Nanocomposites of poly(L-lactide) (PLLA) - magnesium oxide whiskers (MgO) prepared by in-situ polymerization: fabrication, crystalline behaviors, mechanical property and in-vivo degradation" reports about a study on the synthesis and characterization of PLLA-based nanocomposites containing MgO particles, obtained through in-situ polymerization.

Although the topic of the work looks very interesting, I suggest to the authors to intensively revise the whole manuscript and to re-submit it.

In fact, the English language of the manuscript needs to be significantly improved; additionally, some explaination of the obtained results have to be clarified. As an exemple, it is not clear the trend of the thermal and mechanical properties as a function of the content of  MgO particles. The comment about the mechanical properties needs to be rewritten, also considering the results coming from DSC measurements and the degree of dispersion of the particles within polymer matrix. 

Reviewer 2 Report

The aim of this study was to develop a PLLA-MgO composites by one step in-situ polymerization method. Several tests were performed to identify the physical and chemical properties of the composite. The major finding of this research is that the whisker content of 1.0 wt% MgO exhibited the highest molecular weight and enhanced the mechanical properties greatly. In addition, they found that the addition of MgO positively affect the degradation of the PLLA matrix and improve its bioactivity. Overall, the results obtained from this paper are sound and can support their conclusion. To improve the readable of this manuscript, the reviewer has the following comments:

1.      It is not clear the purpose of “Molecular weights (Mη) measurements” experiment. It seems also carried out to test the MW of in vitro samples. If so, please mention it. In Figure 11, the author provided “The weight average molecular weight of PLLA and PLLA1 in the rabbit body.” However, it is not clear how to measure this molecular weight and how to prepare the samples. The author should add more detail description of these.

2.      Regarding the animal experiment, I cannot find the method of anesthesia and sacrifice. In addition, the approval number of the animal experiment should be provided. Please add this information to the next version of the manuscript.

3.      It is hard to understand the Fig.13. What is the relationship of “a , b”and “A, B”? What the red circle (solid and dashed line) demonstrated? “. It was mentioned that “…the whole samples; b. the partial enlargement; the bone cell.” Where is the bone cell? What is the blue arrow to indicate? The authors should revise the figure legend to provide more readable information.

4.      In figure 7a, there is a dramatic difference in DSC read for 0.5% sample but cannot find in 1% and 1.5% samples. Please provide more discussion on this phenomenon.

5.      How to prepare the sample of Fig.1a.

6.      How many samples were tests in each experiment? Especially, the data showed in Fig. 10 demonstrated the error bar of each data. Please add the test number in the revised manuscript.

7.      In this study, a total of 10 rabbits were used. How to divide the group. How many samples in each group? Why the error bar did not provide in Fig. 11. Please perform statistical analysis on this result.